# Antimicrobial Potential of Strontium Hydroxide on Bacteria Associated with Peri-Implantitis

**DOI:** 10.3390/antibiotics10020150

**Published:** 2021-02-03

**Authors:** Hatem Alshammari, Jessica Neilands, Gunnel Svensäter, Andreas Stavropoulos

**Affiliations:** 1Department of Periodontology, Faculty of Odontology, Malmö University, SE-205 06 Malmö, Sweden; hatem.alshammari@mau.se; 2Section for Oral Biology and Pathology, Faculty of Odontology, Malmö University, SE-205 06 Malmö, Sweden; jessica.neilands@mau.se (J.N.); gunnel.svensater@mau.se (G.S.); 3Biofilms—Research Center for Biointerfaces, Malmö University, SE-205 06 Malmö, Sweden; 4Division of Regenerative Dental Medicine and Periodontology, University Dental Clinics, CUMD, University of Geneva, CH-1211 Geneva, Switzerland

**Keywords:** strontium, antimicrobial, peri-implantitis, bacteria

## Abstract

*Background*: Peri-implantitis due to infection of dental implants is a common complication that may cause significant patient morbidity. In this study, we investigated the antimicrobial potential of Sr(OH)_2_ against different bacteria associated with peri-implantitis. *Methods*: The antimicrobial potential of five concentrations of Sr(OH)_2_ (100, 10, 1, 0.1, and 0.01 mM) was assessed with agar diffusion test, minimal inhibitory concentration (MIC), and biofilm viability assays against six bacteria commonly associated with biomaterial infections: *Streptococcus mitis*, *Staphylococcus epidermidis*, *Aggregatibacter actinomycetemcomitans*, *Porphyromonas gingivalis*, *Escherichia coli*, and *Fusobacterium nucleatum*. *Results*: Zones of inhibition were only observed for, 0.01, 0.1, and 1 mM of Sr(OH)_2_ tested against *P. gingivalis*, in the agar diffusion test. Growth inhibition in planktonic cultures was achieved at 10 mM for all species tested (*p* < 0.001). In biofilm viability assay, 10 and 100 mM Sr(OH)_2_ showed potent bactericidal affect against *S. mitis*, *S. epidermidis*, *A. actinomycetemcomitans*, *E. coli*, and *P. gingivalis*. *Conclusions*: The findings of this study indicate that Sr(OH)_2_ has antimicrobial properties against bacteria associated with peri-implantitis.

## 1. Introduction

Dental implants are commonly used to replace missing teeth, with high success- and patient satisfaction rates [1,2,3]. Nevertheless, dental implants often become infected, resulting in the development of peri-implantitis. Peri-implantitis is, similarly to periodontitis, a polymicrobial infection that affects susceptible hosts, and is characterized by inflammation in the peri-implant mucosa and progressive bone resorption [4]; additionally, it may lead to implant loss. Although striking similarities in the oral biofilm established on teeth and implants have been demonstrated [5], recent studies point to significant differences between the microbiome profiles associated with periodontitis and peri-implantitis. For example, by employing the global approach of pyrosequencing to investigate subgingival periodontal and submucosal peri-implant microbial communities, it can be demonstrated that peri-implantitis-associated communities have significantly lower species diversity than those associated with periodontitis [6]. Studies have suggested that some bacterial clusters commonly found in periodontitis lesions, e.g., *Porphyromonas gingivalis* (*P. gingivalis*) and *Aggregatibacter actinomycetemcomitans* (*A. actinomycetemcomitans*) [7,8], but also others uncommon to periodontitis, e.g., *Streptococcus mitis* (*S. mitis*) [9], *Staphylococcus epidermidis* (*S. epidermidis*) [10] and *Staphylococcus aureus* (*S. aureus*) [9,11], may play a significant role in the pathogenesis of peri-implantitis. Further, enterobacterial contamination, especially with *Escherichia coli* (*E. coli*), has also been shown in implants with peri-implantitis [12]. The estimated weighted mean prevalence of peri-implantitis, on a patient level, is 22% [13,14]. In this context, treatment of peri-implantitis is complex, and the various proposed protocols commonly include the use of systemic antibiotics; however, the added benefit of antibiotics—especially on the long-term—is questionable [15,16,17]. Thus, considering the difficulty in effectively treating peri-implantitis, often resulting in multiple/repeated surgical interventions in patients with multiple implants, together with the issue of antibiotic resistance, the development of preventive strategies to control implant-associated infections appears pertinent. One suggested approach is associating substances with antimicrobial properties with the implant surface [18,19]. For example, antimicrobial peptides (AMPs), antibiotics, and metal ions (e.g., silver (Ag), zinc (Zn), and strontium (Sr)) associated with the surface of metallic implants have shown bactericidal properties against different periodontal pathogens in vitro [20]. Specifically, Sr is especially interesting since it has been shown to inhibit the growth of *E. coli*, *S. aureus*, *A. actinomycetemcomitans,* and *P. gingivalis*, while promoting the osteogenic and angiogenic properties of titanium [21] in vitro, and to enhance bone healing and implant osseointegration in vivo [22,23,24].

Thus, the aim of the present study was to further investigate the antimicrobial potential of Sr(OH)_2_ against different bacteria associated with peri-implantitis.

## 2. Materials and Methods

### 2.1. Bacterial Strains and Media

In this study, the following bacterial species were used: *S. epidermidis* ATCC 35984, *E. coli* ATCC 25922, *S. mitis*, *P. gingivalis*, *A. actinomycetemcomitans*, and *Fusobacterium nucleatum* (*F. nucleatum*). *S. mitis* had been isolated supragingivally from a healthy male donor, while *P. gingivalis*, *A. actinomycetemcomitans,* and *F. nucleatum* had all been recovered from subgingival biofilms from patients with established periodontitis. Tryptone yeast extract (TYE) (Becton, Dickinson and Co., Albertslund, Denmark) growth media was used for *S. mitis*, *S. epidermidis*, *A. actinomycetemcomitans*, and *E. coli*; Peptone yeast extract (PYE) (Becton, Dickinson and Co., France) was used for *P. gingivalis*; *F. nucleatum*. *S. mitis*, *S. epidermidis*, *E. coli,* and *A. actinomycetemcomitans* were incubated in 5% CO_2_; while *P. gingivalis* and *F. nucleatum* were incubated anaerobically (10% H_2_, 5% CO_2_ in N_2_). The bacterial isolates were stored at −80 °C and recovered on Brucella agar before experimental use.

### 2.2. Preparation of Strontium Hydroxide Sr(OH)_2_

Different concentrations of strontium hydroxide Sr(OH)_2_ (Sigma-Aldrich Sweden AB, Stockholm, Sweden) were prepared by dissolving the Sr(OH)_2_ in sterile water. A 100 mM Sr(OH)_2_ stock solution was prepared and further diluted to 10, 1, 0.1, and 0.01 mM.

### 2.3. Agar Diffusion Test

Colony-forming units of each strain, grown on Brucella agar, were suspended in 1.8 mL dilution blank, until reaching an optical density of 0.1 at OD 600 nm. A 100 µL-aliquot of each suspension was spread evenly on Brucella agar using glass beads to ensure even growth [25]. Five cylindrical holes (6 mm in diameter) were prepared with a biopsy punch through the entire agar and filled with 80 µL of 10, 1, 0.1, or 0.01 mM Sr(OH)_2_. Agars inoculated with *S. mitis*, *S. epidermidis*, *E. coli*, and *A. actinomycetemcomitans* were incubated at 37 °C in 5% CO_2_, while agar plates with *F. nucleatum* and *P. gingivalis* were incubated in anaerobic environment (10% H_2_, 5% CO_2_ in N_2_) at 37 °C. The growth inhibition zone was evaluated after 2–10 days, depending on the growth rate of the various bacterial species, by using photographs and measuring the distance from two opposite points at the border of the inhibition zone. Each test was made in triplicates, and averages were calculated for each concentration.

### 2.4. Minimal Inhibitory Concentration (MIC)

Suspensions of *S. mitis*, *S. epidermidis*, *E. coli*, and *A. actinomycetemcomitans* were prepared by vortexing colony forming units from Brucella agar in TYE to an OD600 of 0.1. Sr(OH)_2_ was added to the suspensions to give a final concentration of 10, 1, 0.1, 0.01, or 0.001 mM, respectively. Bacterial suspensions were incubated accordingly, and OD600 was evaluated at different timepoints, corresponding to the timepoint of maximum growth of control bacteria. Each test was done in duplicates, and the effect of Sr(OH)_2_ was reported as MIC values at maximum growth. For *P. gingivalis* and *F. nucleatum*, the same procedure was applied with the exception that TYE solution was replaced by PYE and that these were incubated under anaerobic conditions.

### 2.5. Biofilm Viability Assay

Single-species biofilms of *S. mitis*, *S. epidermidis*, *E. coli,* and *A. actinomycetemcomitans* were prepared by adding 120 µL of bacterial suspension prepared as described above for MIC, to each well of an Ibidi mini flowcell (Ibidi^®^ µ-Slide, Ibidi GmbH, Martinsried, Germany) followed by incubation over-night in 5% CO_2_. The hydrophilic surface of the Ibidi promotes bacterial cell adhesion and facilitates confocal scanning laser microscope (CSLM) analysis of these cells after being subjected to Sr(OH)_2_. The following day, the chambers were rinsed twice with 0.9% NaCl, followed by addition of 120 µL Sr(OH)_2_ at 10 and 100 mM concentrations; 0.9% NaCl solution was used as a control. The chambers were incubated for two hours after which the Sr(OH)_2_ was removed. Subsequently, 60 µL of LIVE/DEAD^®^ BacLight™ (Invitrogen, Eugene, USA) was added and the viability of the biofilm cells examined using an Eclipse TE2000 inverted CSLM (Nikon). A total of 10 CSLM images were acquired utilizing the software EZ-C1 v.3.40 build 691 (Nikon) at a resolution of 512 × 512 pixels and with a zoom factor of 1.0, giving a final pixel resolution of 0.42 µm/pixel. The number of viable cells was analyzed manually by counting the number of red and green cells in each image. For *P. gingivalis* and *F. nucleatum*, the same procedure was applied except that they were incubated under anaerobic conditions.

### 2.6. Statistical Analysis

Statistical analysis was done using One-way ANOVA and post-hoc tests (IBM SPSS Statistics 16, SPSS Inc., Chicago, IL, USA) to compare growth inhibition values in the different groups. The level of significance was set at (*p* < 0.01).

## 3. Results

### 3.1. Agar Diffusion Test

To evaluate the effect of Sr(OH)_2_ on bacterial growth, an agar diffusion assay was used. Uniform growth of bacterial colonies was observed on Brucella agar after 24 h for *E. coli* and *S. epidermidis*, 2 days for *S. mitis*, 5 days for *A. actinomycetemcomitans* and *F. nucleatum*, and 8 days for *P. gingivalis*. For *S. mitis*, *S. epidermidis*, *A. actinomycetemcomitans*, *E. coli*, and *F. nucleatum*, low growth inhibition was achieved at all Sr(OH)_2_ concentrations tested as indicated by confluent growth close to the hole filled with Sr(OH)_2_. However, inhibition zones were observed for *P. gingivalis* at 0.01, 0.1, and 1 mM concentrations, with mean diameters of 15, 24, and 21 mm, respectively. Figure 1 shows the inhibition zone observed in 0.1 mM Sr(OH)_2_ against *P. gingivalis* (a), whereas similar effect was not observed for the same concentration but tested against *F. nucleatum* (b). A tendency for Sr(OH)_2_ to precipitate at the bottom of the plate, especially at the higher concentrations (10 and 100 mM), was observed.

### 3.2. Minimal Inhibitory Concentration (MIC)

To further evaluate the effect of Sr(OH)_2_ on the various bacteria included in this study, a growth inhibition test of planktonic cells using different concentrations of Sr(OH)_2_ was performed. Maximum growth of control bacteria was achieved at different time points for the different species, ranging from 12 h to 9 days (Figure 2). Only 10 mM Sr(OH)_2_ exhibited an effect against bacterial growth. Specifically, a 60% growth inhibition was achieved for *E. coli*, while growth of *S. mitis*, *S. epidermidis*, *A. actinomycetemcomitans*, *P. gingivalis*, and *F. nucleatum* was almost completely inhibited. The inhibition of growth by 10 mM Sr(OH)_2_ was statistically significant for all species tested (*p* < 0.001).

### 3.3. Biofilm Viability Assay

To evaluate the effect of Sr(OH)_2_ on biofilm cells, over-night grown single species biofilms were exposed to two different concentrations of Sr(OH)_2_ for two hours. A potent to moderate antimicrobial effect was demonstrated. Specifically, 100 mM of Sr(OH)_2_ killed all tested bacterial strains (Figure 3), while 10 mM Sr(OH)_2_ resulted in less than 1% viable cells for *A. actinomycetemcomitans*, *S. mitis*, and *S. epidermidis*, compared to 86%, 74%, and 84%, respectively, for controls. Further, the use of 10 mM Sr(OH)_2_ resulted in only 8% and 11% viable cells of *E. coli* and *P. gingivalis*, compared to 90% and 25% in the controls, respectively.

Regarding *F. nucleatum*, both the control and Sr(OH)_2_ samples revealed damaged bacterial cells, in repeated tests, most likely due to prolonged exposure of *F. nucleatum* into oxygen during the LIVE/DEAD staining process. 

## 4. Discussion

The results of the present study indicated that Sr(OH)_2_ has antimicrobial properties against several bacteria associated with peri-implantitis. Specifically, exposure to 10 mM Sr(OH)_2_ resulted in complete growth inhibition of *S. mitis*, *S. epidermidis*, and *F. nucleatum*, and in significant growth inhibition for *A. actinomycetemcomitans*, *P. gingivalis*, and *E. coli*, based on OD600 during an MIC assay. Furthermore, 10 mM Sr(OH)_2_ showed bactericidal activity for all the strains tested, with most of bacterial cells being partially or completely damaged in a bacterial viability assay.

The finding that Sr exerts antimicrobial activity is in accordance with previous reports [21,26,27]. For example, Zhou et al. explored the antimicrobial potential of Sr-coated titanium disks against *S. aureus* and *E. coli*, using a bacterial counting method [21]. Sr-coated titanium disks, immersed into a bacterial solution, showed a strong antibacterial potential at 1, 14, and 28 days. Liu et al. [26] investigated the antimicrobial properties of Sr against *A. actinomycetemcomitans* and *P. gingivalis*. The bacteria were incubated under anaerobic conditions, and the effect of different concentrations of Sr (4, 2, 1, 0.5, and 0.25 mM) in a glass particulate suspension in brain heart infusion broth (BHI) was assessed after 2, 4, and 6 h. It was observed that Sr had strong antibacterial effect that increased intensely from 2 to 6 h and was proportional to the amount of Sr released into the solution, i.e., the highest Sr-concentration (4 mM) showed the most prominent effect. Brauer et al. [28] reported similar results against *S. aureus* and *Streptococcus faecalis*, by substituting calcium (Ca) with Sr in bioactive glasses. Using tryptone soya broth (TSB) and counting viable cells in colony forming units, they observed pronounced bacterial growth inhibition of both organisms over 6 days, when Ca was replaced with Sr. In yet another study, by Jayasree et al., substitution of CaCO_3_ with SrCO_3_ in dental cements was associated with a significant antibacterial effect against *S. aureus* and *E. coli* up to 7 days [29]. In the present study, 10 mM Sr(OH)_2_ showed potent antibacterial effect against all species tested, including *P. gingivalis*, *A. actinomycetemcomitans*, and *E.coli*, and had a statistically significant antimicrobial action compared to the control in the growth inhibition assay. Obviously, the wide range of effective concentrations in the above-mentioned studies and the one herein does not allow any conclusion about the minimum effective antimicrobial concentration of Sr, which may, for some of the species investigated herein, lie somewhere between 4 and 10 mM.

In the above-mentioned study of Jayasree et al., the antibacterial effect of SrCO_3_-enhanced dental cements was attributed to the higher pH, which gradually increased overtime, due to the sustained Sr^2+^ ion release up to 6 weeks. Herein, higher concentrations of Sr logically also had higher pH, which, in turn, could explain the antimicrobial effect on the various bacterial strains. Most bacterial species prefer to grow in pH around neutral, and changes in the pH might affect their ability to grow, as well as their viability and properties. For example, it has been demonstrated that bacterial growth inhibition can be achieved through an alkaline environment [30]. This antibacterial effect has been attributed to either induction of oxidative stress (ROS) causing damage to the cell membrane, or interference with metabolic activity through inactivation of adenosine triphosphate (ATP) synthesis.

It the present study, despite several attempts, 100 mM Sr(OH)_2_ was the highest concentration achievable; above this concentration, Sr failed to dissolve properly in the liquid. This concentration was initially diluted 10 times and tested. Then, the five highest concentrations were selected for further testing. In general, the lower concentrations (i.e., < 10 mM) did not have any antimicrobial effect, except in the agar diffusion test where 0.01, 0.1, and 1 mM concentrations were able to induce zones of inhibition only for *P. gingivalis*. In this context, in the agar diffusion test, a negligible antibacterial effect of Sr(OH)_2,_ was observed even with the high concentrations tested. This was most likely due to the fact that Sr—especially the high concentrations—tended to precipitate at the bottom of the agar plates. Furthermore, in the higher concentrations, where the liquid/Sr ratio was smaller than in the lower concentrations, the liquid may have evaporated during the incubation before the Sr was able to diffuse properly into the agar mass. A similar observation has been reported by Li Y et al. [31], where the highest concentration used in their study did not show any zone of inhibition, compared to lower concentrations (25 mol% SrCO_3_ vs. 5 and 10 mol% SrCO_3,_ respectively) that showed the highest zones of inhibition against *S. aureus* in an Agar disk-diffusion test.

In this study, we aimed at investigating the antimicrobial potential of Sr(OH)_2_ against different bacterial species, including Gram-negative and -positive bacterial strains, facultative anaerobes and anaerobic species, early and late colonizers, as well as other bacterial species reported to be associated with peri-implantitis. We employed three different well known techniques in order to get a better picture of the effect of Sr(OH)_2_. Sr(OH)_2_ showed a potent effect in both planktonic cells and biofilm cells. Bacteria within a biofilm have been shown to be more resistant to antimicrobial agents [32,33,34], compared with when grown as planktonic cells; this stresses the importance of using biofilm models. Indeed, a major component in the pathogenesis of peri-implantitis is biofilm formation [35]. Obviously, the results of the present in vitro study cannot translate directly to the clinical situation, where not only the oral biofilm is by far more complex but also the Sr concentration in the local environment most likely varies. Although different Sr(OH)_2_ concentrations were tested herein, each concentration was set “a priori” and was constant for the entire experiment, whereas in the clinical situation with a Sr-loaded device, the concentration is most likely not constant. The concentration of Sr(OH)2 in the local environment depends on how much is dissolved/released from a device, and this in turn depends largely—except from the loading technology and/or device properties—on local fluid dynamics, thus the possibility that if the concentration of Sr in the local environment is too high, mammalian cells may respond similarly to the bacteria should be considered. Therefore, the aspect of Sr toxicity on mammalian cells when Sr is delivered at antimicrobial levels needs to be addressed in future experiments.

## 5. Conclusions

The results of the present study indicated that Sr(OH)_2_ at a concentration of 10 mM interferes with the growth and/or shows bactericidal properties against various oral bacteria associated with peri-implantitis. The results further suggest that Sr coating of implants, abutments, and/or fixation screws may be a relevant strategy for the prevention of peri-implantitis and warrant further investigation.

## Figures and Tables

**Figure 1 antibiotics-10-00150-f001:**
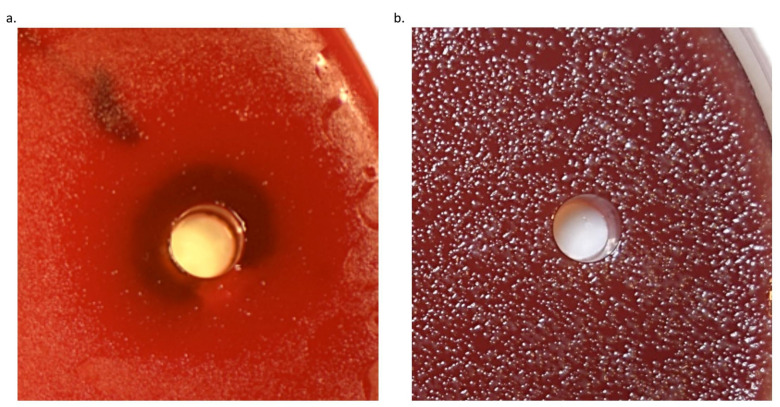
Growth of *P. gingivalis* grown on Brucella agar plate showing inhibition zone associated with 0.1 mM Sr(OH)_2_ (**a**). No zone of inhibition was observed with the same concentration tested against *F. nucleatum* (**b**).

**Figure 2 antibiotics-10-00150-f002:**
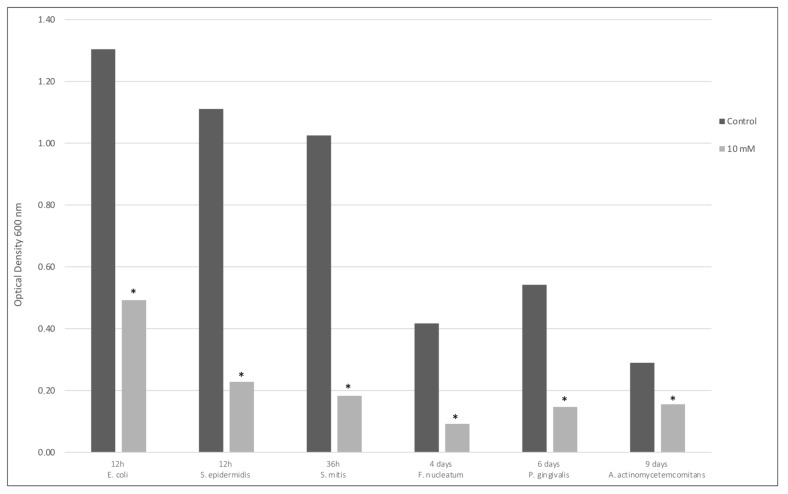
Control and 10 mM Sr(OH)_2_ growth inhibition values at maximum growth of different bacteria at different time points. * *p* < 0.001 compared to control.

**Figure 3 antibiotics-10-00150-f003:**
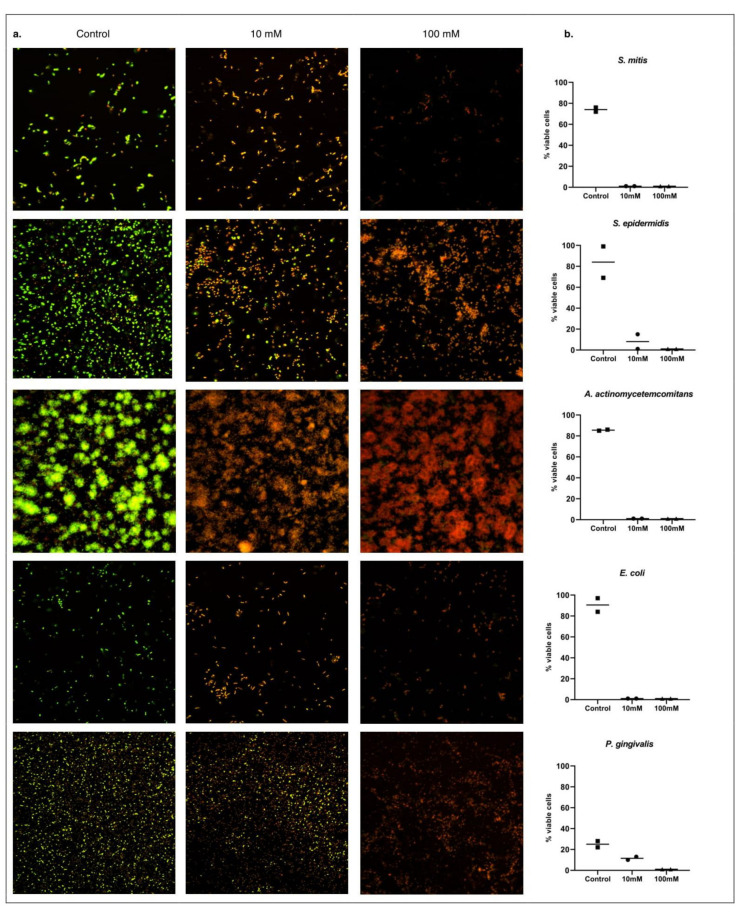
LIVE/DEAD BacLight-stained biofilm cells observed under confocal scanning laser microscope (CSLM); viable cells appear green and dead cells red (**a**). Percentage of viable cells in each experiment (**b**).

## Data Availability

Data available from the authors upon reasonable request.

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
