# Peer review of "Antimicrobial Potential of Strontium Hydroxide on Bacteria Associated with Peri-Implantitis"

_antibiotics, 2021, doi:10.3390/antibiotics10020150_

Round 1
Reviewer 1 Report
The findings of this study indicate that strontium hydroxide has antimicrobial properties against various oral bacteria associated with peri-implantitis. The researchers determined a concentration of 10mM Sr(OH)2 which has bactericidal properties against various oral bacteria. The obtained research results are very interesting and promising.My opinion:
- Interesting research results
- The aspect of practical use of research results fulfilled
The areas of the strength of this paper:
- use of appriopriate research methods and equipment,
- - study on biofilm cells,
- - interesting and practical research results.
The areas of the weakness of this paper:
- why there has been damage too the cells Fusobacterium nucleatum?
Author Response
Thank you very much for your kind comments!
Regarding the point you raised, about the damaged F. nucleatum cells, this was most likely due to prolonged exposure of F nucleatum into oxygen during the dead/live backlight staining process. The following caption is now included in the results section:
Regarding F. nucleatum, both the control and Sr(OH)2 samples revealed damaged bacterial cells, in repeated tests, most likely due to prolonged exposure of F. nucleatum into oxygen during the dead/live staining process.
Furthermore, additional adjustments have been made to the original draft to address points raised by the other reviewers; all changes are highlighted in the submitted manuscript in yellow.
Reviewer 2 Report
The submitted manuscript for publication in Antibiotics describes the toxic effects exerted by different concentration of Sr(OH)2 against different bacteria strains as well as bacteria forming biofilm for treatment of peri-implantitis. The research is well-performed and offers a good novelty in field of the treatment bacteria responsible of oral infections. The English should be extensively revised removing some mistakes on the manuscript and the revision should give a more fluent aspect to the speech. Moreover, the authors should have submitted a paper written according to the Antibiotics template which simplify the reviewer’s revision. In any case, the work can be published in Antibiotics after the indicated modifications and/or corrections:
Abstract: this section needs to be completely renovated and re-written since, in that form, it appears as a list of data. The abstract should highlight the salient aspects of the work and report the most interesting results.
First line of page 2: please add a dot after peri-implantitis.
Please, bacteria strains and any other words needed to be abbreviated should be first cited in main text and then abbreviated. The authors are invited to check all the bacteria strains since a lot of mistakes have been made (e.g. the last part of introduction and 2.1 paragraph)
Paragraph 2.2, please indicate the supplier for Sr(OH)2
Paragraph 3.1: “However, inhibition zones were observed for P. gingivalis”
Figure 1 does not include imagines of inhibition zones at 0.01 and 1 mM. Please, these details should be introduced as Figure 1a. Moreover, figure 1b never appears in the main text.
Paragraph 3.2: “for the rest of the bacterial species”. Here the authors should add the tested bacteria avoiding that readers go up and down through the text.
Discussion: “The results of the present study indicated that Sr(OH)2 has antimicrobial properties against several bacteria associated with peri-implantitis”. And what about the effects of Sr(OH)2 against mammalian cells? could you report any comments and/or test for this kind of activity? It is worth to underline that a suitable agent for treating peri-implantitis cannot show toxic effects against mammalian cells. Please, argue this issue.
“Further, 10 mM Sr(OH)2 showed”. Please replace further with furthermore.
“The finding that Sr exerts antimicrobial activity is in accordance …. a significant antibacterial effect against S. aureus and E. coli up to 7 days [29]”. Please move this part to the introduction with a consequent rearrangement of reference list.
“Sr(OH)2 consists of 1 Sr ion and 2 OH ions”. The authors are strongly invited to correct this sentence. They must only talk of Sr2+ and OH- ions.
“bacterial species including gram negative and positive bacteria strain”
“Biofilm bacteria”. Please replace with “bacteria forming biofilm”
“..the results herein suggest that Sr coating of implants, abutments, and/or fixation screws may be a relevant strategy for the prevention of peri-implantitis and warrant further investigation”. Please, move this sentence to the conclusion.
Please, correct the references list according to MDPI template.
Author Response
Thank you very much for your kind comments and insightful and constructive review. We believe the manuscript has improved based on your criticism/suggestions.
Please find below a point-by-point response in your comments (all changes in the manuscript are highlighted):
- The English should be extensively revised removing some mistakes on the manuscript and the revision should give a more fluent aspect to the speech. Moreover, the authors should have submitted a paper written according to the Antibiotics template which simplify the reviewer’s revision.
Our response: We have revised the manuscript removing spelling mistakes and trying to have a more fluent read. All changes are highlighted (in yellow) in the resubmitted manuscript.
- Abstract: this section needs to be completely renovated and re-written since, in that form, it appears as a list of data. The abstract should highlight the salient aspects of the work and report the most interesting results.
Our response: the abstract was largely rewritten to highlight important aspects of the work and report the most interesting results.
- First line of page 2: please add a dot after peri-implantitis.
Our response: done.
- Please, bacteria strains and any other words needed to be abbreviated should be first cited in main text and then abbreviated. The authors are invited to check all the bacteria strains since a lot of mistakes have been made (e.g. the last part of introduction and 2.1 paragraph)
Our response: Mistakes were identified and adjusted (highlighted in the resubmitted manuscript).
- Paragraph 2.2, please indicate the supplier for Sr(OH)2
Our response: name of supplier company included.
- Paragraph 3.1: “However, inhibition zones wereobserved for gingivalis”
Our response: done.
- Figure 1 does not include imagines of inhibition zones at 0.01 and 1 mM. Please, these details should be introduced as Figure 1a. Moreover, figure 1b never appears in the main text.
Our response: Elaboration on figure 1 was added to the main text (section: 3.1. agar diffusion test). The added section is highlighted in the text.
- Paragraph 3.2: “for the rest of the bacterial species”. Here the authors should add the tested bacteria avoiding that readers go up and down through the text.
Our response: the sentence changed including the names of the tested bacteria.
- Discussion: “The results of the present study indicated that Sr(OH)2 has antimicrobial properties against several bacteria associated with peri-implantitis”. And what about the effects of Sr(OH)2 against mammalian cells? could you report any comments and/or test for this kind of activity? It is worth to underline that a suitable agent for treating peri-implantitis cannot show toxic effects against mammalian cells.Please, argue this issue.
Our response: Thank you for this important comment! We have added the following caption at the end of the discussion to address this point:
The concentration of Sr(OH)2 in the local environment depends on how much is dissolved/released from a device, and this in turn depends largely – except from the loading technology and/or device properties – on local fluid dynamics; thus, the possibility that if the concentration of Sr in the local environment is too high, mammalian cells may respond similarly to the bacteria should be considered. Therefore, the aspect of Sr toxicity on mammalian cells when delivered at antimicrobial levels needs to be addressed in future experiments.
- “Further, 10 mM Sr(OH)2 showed”. Please replace further with furthermore.
Our response: done.
- “The finding that Sr exerts antimicrobial activity is in accordance …. a significant antibacterial effect against S. aureus and E. coli up to 7 days [29]”. Please move this part to the introduction with a consequent rearrangement of reference list.
Our response: Thank you for this suggestion. In the introduction (page 2, second paragraph) we point to previous reports about the antimicrobial activity of Sr and refer to some important references. However, in the discussion, we address this issue in more detail and in relation to current findings herein. As such, we would very much prefer to keep this part as is in the discussion. We hope this is OK with you.
- “Sr(OH)2 consists of 1 Sr ion and 2 OH ions”. The authors are strongly invited to correct this sentence. They must only talk of Sr2+ and OH-
Our response: the sentence was rephrased as suggested.
- “bacterial species including gram negative and positive bacteriastrain”
Our response: done.
- “Biofilm bacteria”. Please replace with “bacteria forming biofilm”
Our response: done.
- “..the results herein suggest that Sr coating of implants, abutments, and/or fixation screws may be a relevant strategy for the prevention of peri-implantitis and warrant further investigation”. Please, move this sentence to the conclusion.
Our response: the sentence was moved to conclusion.
- Please, correct the references list according to MDPI template.
Our response: We apologize for any misunderstanding, but we have downloaded the reference style of the journal from the website (http://endnote.com/downloads/style/mdpi), added to endnote 9, and the reference list was generated accordingly.
Reviewer 3 Report
Interesting and well written article.
Methods are appropriate. Results are clearly presented.
Discussion is interesting
Author Response
Thank you very much for your kind comments!
For your information, we have revised the manuscript removing spelling mistakes and trying to have a more fluent read, as it was requested by another reviewer. All changes are highlighted (in yellow) in the resubmitted manuscript.
Round 2
Reviewer 2 Report
the authors carefully revised their manuscript addressing all the raised issues and providing a point-by- point cover letter.
I have no further comments and I recommend this article for publication.
Regards